# Improving the Annotation of the Venom Gland Transcriptome of *Pamphobeteus verdolaga*, Prospecting Novel Bioactive Peptides

**DOI:** 10.3390/toxins14060408

**Published:** 2022-06-15

**Authors:** Cristian Salinas-Restrepo, Elizabeth Misas, Sebastian Estrada-Gómez, Juan Carlos Quintana-Castillo, Fanny Guzman, Juan C. Calderón, Marco A. Giraldo, Cesar Segura

**Affiliations:** 1Grupo Toxinología, Alternativas Terapéuticas y Alimentarias, Facultad de Ciencias Farmacéuticas y Alimentarias, Universidad de Antioquia, Medellín 050012, Colombia; cristian.salinas@udea.edu.co (C.S.-R.); sebastian.estrada@udea.edu.co (S.E.-G.); 2Corporación para Investigaciones Biológicas, Medellín 050012, Colombia; elizabeth.misas@gmail.com; 3Centro de Investigación en Recursos Naturales y Sustentabilidad, Universidad Bernardo O’Higgins, Aven-ida Viel 1497, Santiago 7750000, Chile; 4Facultad de Medicina, Universidad Cooperativa de Colombia, Medellín 050012, Colombia; juan.quintanac@ucc.edu.co; 5Núcleo Biotecnología Curauma (NBC), Pontifícia Universidad Católica de Valparaíso, Valparaíso 2374631, Chile; fanny.guzman@pucv.cl; 6Physiology and Biochemistry Research Group-PHYSIS, Faculty of Medicine, University of Antioquia, Medellín 050012, Colombia; jcalderonv00@yahoo.com; 7Biophysics Group, Institute of Physics, University of Antioquia, Medellín 050012, Colombia; mantonio.giraldo@udea.edu.co; 8Grupo Malaria, Facultad de Medicina, Universidad de Antioquia, Medellín 050012, Colombia

**Keywords:** *Pamphobeteus verdolaga*, spider, tarantula, transcriptomic, theraphosid, peptide prospection, venom gland, toxin, annotation, non-model organism

## Abstract

Spider venoms constitute a trove of novel peptides with biotechnological interest. Paucity of next-generation-sequencing (NGS) data generation has led to a description of less than 1% of these peptides. Increasing evidence supports the underestimation of the assembled genes a single transcriptome assembler can predict. Here, the transcriptome of the venom gland of the spider *Pamphobeteus verdolaga* was re-assembled, using three free access algorithms, Trinity, SOAPdenovo-Trans, and SPAdes, to obtain a more complete annotation. Assembler’s performance was evaluated by contig number, N50, read representation on the assembly, and BUSCO’s terms retrieval against the arthropod dataset. Out of all the assembled sequences with all software, 39.26% were common between the three assemblers, and 27.88% were uniquely assembled by Trinity, while 27.65% were uniquely assembled by SPAdes. The non-redundant merging of all three assemblies’ output permitted the annotation of 9232 sequences, which was 23% more when compared to each software and 28% more when compared to the previous *P. verdolaga* annotation; moreover, the description of 65 novel theraphotoxins was possible. In the generation of data for non-model organisms, as well as in the search for novel peptides with biotechnological interest, it is highly recommended to employ at least two different transcriptome assemblers.

## 1. Introduction

Spiders are the venomous animals with the highest number of species reported, with more than 49,000 taxa reported to date [1]. Their highly complex venom, which contains up to 200 different peptides and therefore over 10 million potential bioactive molecules for the whole order, is comprised of low (<1 kDa), medium (1–10 kDa; disulfide bridged peptides (DBPs) or alpha helicoidal peptides (non-disulfide bridged peptides or NDBPs)), and high molecular weight (>10 kDa) peptides and proteins [2,3]. The most abundant components in the venom, the neurotoxic (DBPs) and cytolytic/antimicrobial peptides (NDBPs), show much promise in the development of new tools for research and for applications in agriculture and medicine due to interesting attributes, such as insecticidal activity, modulation of K^+^, Ca^2+^, or Cl^−^ ion channels, and anti-inflammatory and anti-cancer properties [4,5,6,7], as well as bactericidal or bacteriostatic activity found in antimicrobial peptides (AMPs) [8,9].

The secondary structure of DBPs, predominantly formed by β-sheets that are stabilized by 5–10 disulfide bridges, is known as the inhibitory cysteine motif (ICK) [10]. Inside the ICK, two other motifs can be identified: the primary structural motif (PSM) and the extra structural motif (ESM), which are characterized by bridges in the form C1-X(4,6)-C2-X(4,9)-C3-C4-X(2,10)-C5-X(3,14)-C6-X(1,16) and C7-X-C8, respectively (where X is any number of amino acids). The presence of the PSM is sufficient to assume that the peptide has the ability to bind ion channels [4]. Peptides such as Huentoxin-XVI (*Ornithoctonus huwena*) and ProTx-II (*Thrixopelma pruriens*) interrupt nociception in various models, while peptides such as Phα1β and PnPP-19 from the venom of *Phoneutria nigriventer* are around 180 times more potent than ziconotide [11,12,13]. Moreover, DBPs have shown potential in the treatment of other pathologies, such as in models of ischemic stroke as shown by the peptide PcTx1 (*Psalmopoeus cambridgei*), in epilepsy models as shown by the peptide Hm3a (*Heteroscrodra maculate*), in Alzheimer’s disease models as shown by the peptide PhKv (*Phoneutria nigriventer*), and in Parkinson’s disease models as shown by Guangxitoxin-1E toxin [12]. The peptide Gomesin from *Acanthoscurria gomesiana* also shows activity against Gram-positive and Gram-negative bacteria as well as fungi, yeast, and parasites of the genus *Plasmodium* or *Leishmania*, while retaining anti-cancerous activity in a murine melanoma model [14].

On the other hand, the peptides termed NDBPs present a predominant α-helix secondary structure with an abundance of cationic amino acids, such as lysine and arginine, and hydrophobic amino acids, such as leucine, isoleucine, and valine. Being cationic and amphipathic, these peptides prefer to interact with the partially anionic phosphatidyl glycerol and cardiolipin enriched bacterial membranes. Interestingly, transformed tumor cells, rich in phosphatidylserine, are targeted by NDBPs, giving the latter antitumoral and antimicrobial properties [7,15]. Four major families of NDBPs can be described in spiders: lycotoxins, latarcins, cupeinnins, and oxyopinins [15]. Most of these toxins display antimicrobial activities in the millimolar range. However, some toxins display activity against microorganisms of clinical interest in the micromolar range: the peptide lycotoxin I from spiders of the genera *Lycosa* displays activity against methicillin-resistant *Staphylococcus aureus* (MRSA), and the peptide CIT1a from *Lachesana tarabaevi* displays activity against the yeast *Chlamydia trachomatis* [16]. Furthermore, the peptide lycosin-I (from *Lycosa singnoriensis*) exhibits activity against fungi from the genera *Penicillium* and *Aspegillus* and is also able to induce the apoptosis of prostate cancer cells, while the peptide latarcin-3a from *Lachesana tarabaevi* is able to generate pores in the envelope of the HIV virus [12]. 

This broad range of activities shows the enormous biotechnological potential of spider venoms; yet, less than 1% of the hypothetical potential novel molecules have been reported in the literature [17]. Proteomic studies from spider venoms have been limited in the amount of data that they can provide, since the collection of the sample is technically complex [18]. Therefore, transcriptomics appears as a preferable tool for the prospection of bioactive molecules from arachnids, as the collection of the sample is greatly simplified [19,20]. Next-generation-sequencing (NGS) tools allow the collection of vast amounts of data from tissues such as the venom gland. Identification of sequences from NGS data is usually carried by pipelines that rely on alignment methodologies, e.g., BLAST [19,21,22,23]. However, alignment methodologies specialize in the detection of homology between closely related sequences. Thus, for non-model organisms, methodologies for the detection of distantly related sequences, which rely instead on the identification of conserved motifs by statistical models, e.g., hidden Markov models (HMMs) [24,25], must also be implemented, as these could show higher specificity and sensibility. For organisms such as those found in the *Araneae* order, due to the lack of reported data at the genome and transcriptome level, the assembly and the annotation of genomes or transcriptomes become important endeavors in the characterization process; thus, the retrieval of the nucleotide information is paramount. Venoms are highly complex substances: a high splicing activity [26,27], as well as the presence of several duplicated genes within the venom gland [28], accounts for the presence of highly homologous and paralogous sequences that can distort the quantity and quality of the information recovered. Therefore, the use of various assemblers may also be recommended since taxa-specific as well as tissue-specific biases and issues may arise [29,30].

The main objective of this study is to improve the available transcriptomic resources of the spider *Pamphobeteus verdolaga* (*Araneae*: *Theraphosidae*). Previous works by Estrada and collaborators [18,31,32,33] allowed the partial characterization of the venom as well as the identification of over 7137 non-redundant (nr) open reading frames (ORFs) (over 100 sequences haven been disclosed so far on the European Nucleotide Archive under accession number HAHO01), 256 disulfide bridged peptide toxins, and over 45 high molecular mass proteins from the venom gland transcriptome of *P. verdolaga*. Since there is evidence that a single assembly method does not retrieve all genes present within a transcriptome [29,30,34], and there is an expressed need for the generation of transcriptomic data from organisms of the *Araneae* order, we have improved the amount of information obtained from *P. verdolaga’s* venom gland transcriptome by re-assembling the reads with three different free de novo assembly algorithms. The new results allowed us to annotate and report 9232 genes and unveiled new bioactive peptides with potential interest in biomedicine.

## 2. Results

### 2.1. Quality of the Transcripts Assembled with Trinity and SPAdes Are Similar, While Outperforming That of SOAPdenovo-Trans

The 24 million 101 bp pair-ended raw reads of the venom gland transcriptome of *P. verdolaga* were cleaned from low-quality and Illumina adaptor sequences with the software TrimGalore (v0.6.3 available at https://github.com/FelixKrueger/TrimGalore, accessed on 31 July 2020) and assembled using three free access software: Trinity (v2.1.1 available at https://github.com/trinityrnaseq/trinityrnaseq, accessed on 31 July 2020), SPAdes (v3.13.1 available at https://github.com/ablab/spades, accessed on 31 July 2020), and SOAPdenovo-Trans (v1.0.4 available at https://github.com/aquaskyline/SOAPdenovo-Trans, accessed on 31 July 2020). On average, 36% and 85% of the obtained contigs from SPAdes and SOAPdenovo-Trans had less than 200 bp, respectively (Table 1). Removal of these sequences showed that Trinity assembled 21% more contigs than SPAdes and 52% more contigs than SOAPdenovo-Trans on average; however, the N50 metric was higher on average for those contigs assembled with SPAdes (802 bp), followed then by Trinity (784 bp), and finally by SOAPdenovo-Trans (486 bp); the GC content statistic showed no major differences (Table 1). Raw read representation on the assembled contigs was also calculated with the software Bowtie2 (v.2.2.5 available at https://github.com/BenLangmead/bowtie2, accessed 31 July 2020). All assemblies obtained with both Trinity and SPAdes were found to be of high quality, since over 94% of the raw reads were traced back to the assembled contigs (Table 1). For SOAPdenovo-Trans, only the assembly of k-mer 63 was found to be of high quality, as 92% of the raw reads aligned to the assembled contigs (Table 1). A secondary approximation to read representation was carried out using the genome of the common house spider *Parasteatoda tepitadorium* as a template: Trinity’s aligned contigs had an average length higher than that of SPAdes and SOAPdenovo-Trans by 31% and 107%, respectively, and for the total coverage statistic, Trinity’s coverage was found to be 65% larger than that of SPAdes and up to six times larger than that of SOAPdenovo-Trans (Table 1). However, the max alignment length, as well as alignment quality, was the same for all assemblies obtained for Trinity and SPAdes (Table 1).

Assembly completeness was assessed by the recovery of single copy orthologs that are present across Arthropoda with the software BUSCO (v4.1.2 available at https://github.com/WenchaoLin/BUSCO-Mod, accessed on 31 July 2020). In sequences smaller than 200 bp, no complete BUSCO terms were found; however, on average, 3% and 16% of the dataset were observed in fragmented form in sequences coming from the SPAdes and SOAPdenovo-Trans assemblies, respectively (Table 1). For sequences higher than 200 bp, the number of complete BUSCO terms identified was close between Trinity and SPAdes, with 78% in the former and 76% in the latter. In total, 8% and 2% of the retrieved BUSCO terms for SPAdes and SOAPdenovo-Trans, respectively, were duplicated, being three to ten times smaller than the duplicated average number found on the Trinity output, i.e., 24% (Table 1), suggesting that an important percentage of the sequences assembled by Trinity is redundant.

Currently, there is no consensus regarding the ideal set of parameter values that a de novo transcriptome assembly should attain in both basic alignment quality and assembly completeness for it to be qualified as a good assembly. However, when comparing various assemblies, higher N50 values and higher read representation percentages are preferred. Likewise, in a BUSCO analysis, a high percentage of complete terms is preferable; nonetheless, as gene expression profiles are usually unknown in these types of samples, a lower number of missing BUSCO terms is also significant. Therefore, the assemblies that yielded the best overall performance for each software were (i) Trinity k-mer 25, (ii) SPAdes k-mer 31, and (iii) SOAPdenovo-Trans k-mer 63 (Table 1).

### 2.2. Trinity, SPAdes, and SOAPdenovo-Trans Assemblies Show Differences in the Quantity of Associated ORFs and the Functionality of the Respective Annotated Genes in the Transcriptome of P. verdolaga

The assemblies with the best overall performance for each software were structurally annotated (ORF prediction) with the software Augustus (v3.3.3 available at https://github.com/Gaius-Augustus/Augustus, accessed on 31 July 2020), using the *P. tepitadorium* genome as a template for the Generalized Hidden Markov Models (GHMMs). Analysis of only the longest complete ORFs showed that the biggest number of predicted genes was obtained for Trinity, followed by SPAdes, and lastly SOAPdenovo-Trans; the average and maximum ORF length followed the same ranking (Table 2). Due to the percentage of duplicated genes observed in the BUSCO analysis, all ORFs were subjected to a redundancy test with the software CD-HIT (v4.8.1 available at https://github.com/weizhongli/cdhit, accessed on 31 July 2020) using an identity cap of 85%. In total, 4461 of the structurally annotated genes found within the Trinity assembly were redundant, while 1952 and only six of the genes found within the SPAdes and SOAPdenovo-Trans assemblies, respectively, were redundant (Table 2 and Appendix A). A second redundancy analysis, carried out with an identity cap of 85% among all three non-redundant structurally annotated datasets, showed that 8132 of the genes assembled by Trinity were found on either the SPAdes or SOAPdenovo-Trans assembly, 8820 of the genes assembled by SPAdes were found on either the Trinity or SOAPdenovo-Trans assembly, and 7517 of the genes assembled by SOAPdenovo-Trans were found on either the Trinity or SPAdes assembly (Table 2); thus, a total of 7199 sequences were identified as common between all three assemblies (Appendix A). Furthermore, Trinity obtained the highest number of uniquely assembled genes, and SPAdes obtained the highest number of nr ORFs (Table 2).

The non-redundant predicted ORFs for each assembler were functionally annotated with the assistance of the UniProtKB database (Swiss-Prot) (Version 2020_06, available at https://www.uniprot.org, accessed on 31 July 2020) and the InterProScan (IPS) database (Version 80.0, available at https://www.ebi.ac.uk/interpro/, accessed on 31 July 2020) in the software OmicsBox (v2.0.36, BioBam Bioiformatics S.L., Valencia, Spain). A total of 100% of the ORFs for all software obtained InterProScan hits; however, only an average of 56% of the ORFs obtained BLAST hits, which hindered the total number of annotated genes. The highest number of annotated ORFs came from the Trinity assembly, with a total of 7478 sequences (57% of annotated ORFs), followed by SPAdes with 7019 sequences (51% of annotated ORFs), and lasty by SOAPdenovo-Trans with 5134 sequences (61% of annotated ORFs) (Figure 1).

The annotated proteins from each assembly were compared in function of their identified activities, localizations, and associated biological processes. Analysis of the gene ontology terms (GO-terms) allowed: (i) identification of 29 non-redundant GO-terms related to molecular function (activity), of which 95% were shared between Trinity and SPAdes, and only 3% was shared between the three software (Table 3 and Figure 2a), (ii) identification of 33 nr GO-terms related to cellular component (localization), of which 97% were shared between the three assemblies (Table 3 and Figure 2b), and (iii) identification of 113 nr GO-terms related to biological processes, of which 82%, 90%, and 88% were associated to the SOAPdenovo-Trans, SPAdes, and Trinity annotations, respectively (Table 3 and Figure 3). Among the three categories, Trinity obtained the highest number of genes associated to GO-terms, followed closely by SPAdes (Table 3); however, SOAPdenovo-Trans showed the highest number of nr unique GO-terms with a total number of 15 terms, followed by SPAdes with seven terms and Trinity with three terms (Appendix A).

### 2.3. Merging of the Common and Unique Assembled ORFs for Each Software Increases the Annotation Performance

The unique and common sequences found within all three software assemblies (Table 2 and Appendix A) were merged into a single file comprised of 18,338 non-redundant ORFs. These sequences were annotated as described previously for testing if differences in the functionalities of the annotated proteins for each individual software were mitigated. Again, the final number of annotated proteins was heavily influenced by a low number of BLAST hits against the UniProtKB. Still, a total of 9232 ORFs were annotated, implying an increase of 23%, 31%, and 80% more annotated sequences when compared to the individual Trinity, SPAdes, and SOAPdenovo-Trans annotations, respectively (Figure 4 and Appendix A). The distribution of the annotated proteins was 3.05% of the unique SOAPdenovo-Trans sequences, 18.36% of the unique SPAdes sequences, 29.28% of the unique Trinity sequences, and 49.31% of the sequences classified as common between the three assemblies (Table 4).

The GO-terms from the individual software and merged annotations were compared. An increase of 30% in the number of genes associated to molecular function terms was observed in the merge when compared to the Trinity or SPAdes annotations, although only 19 out of the total 29 terms were identified (Table 5), and one common GO-term shared between all individual assemblies was not represented (Appendix A). In total, 5%, 94%, and 89% of the merged molecular function GO-terms were shared with the SOAPdenovo-Trans, SPAdes, and Trinity annotations, respectively (Appendix A). Possible loss of representation was observed in the merged cellular component GO-terms analysis, since a term common to all the individual annotations was not represented within the merge (Appendix A), accompanied by a decrease of 2% in the number of associated genes when compared to the Trinity annotation (Table 5). A total of 97% of the cellular component terms found within the merged annotation are shared between the SOAPdenovo-Trans and SPAdes annotation, and 100% are shared with the Trinity annotation (Appendix A). For biological processes terms, an increase of 7% in the number of associated genes was observed in the merge when compared to the Trinity annotation, as well as five new unique terms, which increased the total number of biological process terms to 118 (Table 5). In total, 97 out of the 118 terms were identified within the merge, with a total of 12 terms present within at least two of the three individual annotations missing (Table 5). However, all uniquely identified terms within the merge showed redundancy to at least one of the 12 missing terms, leaving a total of seven of these without representation (Table 5). In total, 82%, 92%, and 90% of the merged biological process terms were found either on the SOAPdenovo-Trans, SPAdes, and Trinity annotations, respectively (Appendix A).

### 2.4. The P. verdolaga Transcriptome Is Rich in Proteins of Biotechnological Interest

The individual software and merged file ORFs were screened for proteins and peptides of biological interest using a pairwise alignment methodology with BLAST and an HMM methodology with the software hmmcompete (Available at https://github.com/koualab/hmmcompete, accessed on 31 May 2022). The former methodology utilized sequences from the ArachnoServer database (Adb) (UniProtKB—ArachnoServer, version 2022_02, accessed on 31 May 2022) and the *Araneae* subset of the Animal Toxin Annotation Project (ToxProt) (UniProtKB-ToxProt, version 2022_02, accessed on 31 May 2022) as templates, while the latter methodology used the default software’s HMMs. For the BLAST strategy, 150, 139, and 79 ORFs of interest were found within the Trinity, SPAdes, and SOAPdenovo-Trans assemblies, respectively, against the Adb, while 178, 153, and 100 ORFs were observed for Trinity, SPAdes, and SOAPdenovo-Trans, respectively, against ToxProt (Table 6). For Adb, 43% of the hits were related to presynaptic neurotoxins, 15% to proteases, and 13% to lectins; no hits were related to the keywords antiparasitic and kinin (Table 6). For ToxProt, 49% of the hits were related to neurotoxins, 33% to presynaptic neurotoxins, and 6% to protease inhibitors; no sequences were related to the keywords antiparasitic, antiarrhythmic, cytolytic, hyaluronidase, kinin, neurotransmitter hydrolysis, and protease (Table 6). A homology search against the merged file showed the highest number of sequences of interest per keyword, with a total of 174 ORFs for Adb and 220 for ToxProt (Table 6). The HMM strategy allowed the identification of 286, 242, and 162 ORFs of interest for Trinity, SPAdes, and SOAPdenovo-Trans, respectively. In total, 78% of the hits belonged to HMMs related to venom proteins, 19% to venom neurotoxins, and 3% to venom cationic peptides (Table 7). As observed in the BLAST analysis, the merge prospection showed the highest total number of ORFs of interest; however, a smaller number of HMMs related to small cationic peptides were observed when compared to the SPAdes assembly (Table 7).

The potential peptides/proteins of interest within the merged annotation were further analyzed. In total, 146 and 116 ORFs were obtained from the BLAST’s Adb and ToxProt predictions, respectively, since several sequences were related to multiple activities; from the 262 ORFs, 97 were redundant. The 318 ORFs of interest from the HMM analysis were unique; however, 115 ORFs were present in both BLAST and HMM strategies. Finally, a total of 328 ORFs of interest were obtained since 17 and 23 sequences from the BLAST and HMM strategies, respectively, were reported previously [31,33]. As per the HMM classification suggested by Koua and Kuhn-Nentwig [35], 77 families were identified from the peptides/proteins of interest, of which 12% belonged to the α-latrocrustotoxin family, 9% to the leucine-rich peptide family, 9% to the disulfide isomerase family, 8% to the serine protease family, 7% to the tachylectin 5A family, 5% to the α-latroinsectotoxin family, 5% to the α-latrotoxin family, and 3% to the δ-latroinsectotoxin family (Appendix A). All sequences were subjected to mature peptide prediction using the SignalP 6.0 (https://services.healthtech.dtu.dk/service.php?ProP-1.0, accessed on 31 May 2022)) and Phobius (https://phobius.sbc.su.se, accessed on 31 May 2022) software for signal peptide prediction, in combination with the ConoPrec (http://www.conoserver.org/?page=conoprec, accessed on 31 May 2022) and ProP 1.0 (https://services.healthtech.dtu.dk/service.php?ProP-1.0, accessed on 31 May 2022) software for pro-peptide cleavage site prediction. In total, 20% of the ORFs of interest showed presence of both signal peptide and pro-peptide sequences (Appendix A), 19% showed presence of only signal peptide sequences (Appendix A), 47% showed presence of only pro-peptide sequences (Appendix A), and 14% showed neither (Appendix A). The 65 mature sequences were clustered with CD-HIT in 12 groups that contained 74% of the toxins (Appendix A) and were named U-theraphotoxin-Pv5a to U-theraphotoxin-Pv61a, as proposed by King [36] (Appendix A). For the remaining 263 hypothetical toxins, the complete precursor sequence was reported (Appendix A).

## 3. Discussion

Humanity faces many challenges in the search for novel active molecules that help solve animal and human health problems, as well as environmental issues. Recombinant DNA technology and optimization of solid-phase chemical synthesis have allowed peptides and proteins to become feasible and scalable solutions to these problems. Peptides entering clinical trials are now common [37], and venom-derived products are becoming increasingly available [3,38]. Spiders, being the venomous animal with the greatest number of species described to date, are playing a leading role in the prospection of bioactive molecules [1,17]. The de novo assembly of transcriptomes has become a preferred tool for the molecular characterization of non-model organisms due to the unavailability of biological samples [39,40]. However, various challenges, such as the assembly of chimeras, under representation of isoforms, presence of partial transcripts, and biases in highly and lowly expressed genes must still be surmounted [40,41]. Furthermore, evidence suggests that assembly algorithms might distort the assembly of certain types of proteins and perform differently depending on tissues and/or organisms [30,39,41,42]. Therefore, as each piece of information obtained can be crucial in the characterization of these non-model organisms, the use of more than one assembly software must be encouraged. 

### 3.1. The Quality of the Assemblies

All assemblies obtained contig and N50 values in agreement with reported spider transcriptomes, which are highly variable and contain from 5000 up to 201,000 unigenes and N50 values close to 600 bp [43]; this is also in agreement with the previous *P. verdolaga* annotation [32]. Changes in the k-mer length showed only drastic changes for the SOAPdenovo-Trans software, as contigs from the k-mer 31 assembly accounted only for 16% of the raw *P. verdolaga’s* venom gland reads (Table 1), probably due to the tendency of the SOAPdenovo-Trans software to assemble a high proportion of small contigs, as shown in other datasets [42]. For the Trinity and SPAdes software, the smaller k-mer number favored slightly higher numbers of contigs, N50 values, and higher raw read representation within the assembled contigs (Table 1), as previously reported for SPAdes in the assembly of venom gland transcriptomes for snakes of the *Crotalus* genus as well as scorpions of the *Centruroides* and *Hadrurus* genera [29]. Equal max and min alignment length, as well as similar average alignment quality, was observed for all three software assemblies when aligned to the genome of *P. tepitadorium*. However, average alignment lengths of at least one order of magnitude higher than the average N50 value for each software implied the inclusion of gaps, which may have contributed in the reduction of the alignment quality, as it has been observed before in other datasets [44]. As alignment quality metrics alone fail to describe the biological relevance of the assembled transcripts, an analysis of completeness must be carried out with BUSCO. Since values over 60% of complete BUSCO terms are recommended [45], the Trinity and SPAdes assemblies showed good performance while SOAPdenovo-Trans performance was average (Table 1). Nonetheless, levels of missing BUSCO terms between 60 and 10% can be observed within the venom gland transcriptome of snakes and scorpions [29].

### 3.2. Structural and Functional Annotation and the Differences Observed between the Three Software

The ORF prediction used the genome of *P. tepitadorium* as a template for the GHMM model creation. Although the taxa *Theraphosidae* (*Mygalomorphae*) and *Theriidae* (*Araneomorphae*) are estimated to have diverged between 240–300 million years ago [43,46], *P. tepitadorium* is gaining relevance as the center of various molecular, environmental, and developmental models, which virtually make *P. tepitadorium* the “best model spider” described to date [47]. In terms of the total number of ORFs predicted for our dataset, the SPAdes assembly showed a similar performance to that of Trinity, as reported previously [30]; even more so, as about a quarter of the predicted Trinity ORFs were redundant as suggested by the BUSCO analysis (Table 1 and Table 2). Prediction of OFRs in the first annotation of the *P. verdolaga* transcriptome was also comparable, as a total of 16,042 ORFs were predicted with the software TransDecoder [32]. It has been reported that biases in the prediction of certain types of proteins exist within de novo assemblers software [29,30]. For our dataset, this tendency was observed early, as the CD-HIT analysis between Trinity, SPAdes, and SOAPdenovo-Trans showed that the quantity of the common ORFs shared rivaled that of the quantity of the uniquely predicted ORFs (Table 2).

Over 50% of the predicted ORFs were not functionally annotated by the software OmicsBox. Various factors might have contributed to these results: (i) there is low abundance of spider-specific genes in databases such as the UniProtKB [48,49], (ii) phylogenomic analysis of new world theraphosids places them as a separate clade group from other arachnids [50], and (iii) up to 20% of genes within spider transcriptomes are not present within other taxa due to lineage-specific environmental adaptations, while showing the presence of signal peptide, IPS, or Pfam hits [49,51]. This is relevant for our dataset, as 45.5% of the non-annotated sequences have IPS hits that provide insight into the possible function of these genes and might classify them as probable “orphan genes”. Low annotation percentages are also observed in sequences from spiders of clinical importance, such as those from the *Loxoceles* and *Latrodectus* genera, since only 39 to 54% of the described proteins from transcriptomes in these species are correctly identified [52,53]. Comparison to the previous *P. verdolaga* annotation shows that only 45% of the 16,042 ORFs, or 7173 non-redundant genes, were successfully annotated with the software Trinotate [32]. The 4% improvement between both Trinity annotations might be attributable to random artifacts within each assembly or to a slight increase in the number of related sequences within the UniProtKB database between both annotation periods.

Further characterization of the annotated proteins was carried out by analyzing their associated GO-terms with the help of the REVIGO software. In summary: (i) Trinity obtained the highest number of genes associated to all three categories, (ii) SPAdes obtained the overall highest representation of GO-terms in all three categories, and (iii) SOAPdenovo-Trans and SPAdes obtained the highest number of uniquely represented GO-terms in all categories (Table 3). A broad analysis of the keywords associated to the uniquely represented GO-terms showed that for our dataset SOAPdenovo-Trans might favor the prediction of nitrogenated-bases binding proteins, proteins associated to the regulation of cell cycle, and intracellular localization processes (Appendix A). SPAdes might favor the prediction of proteins with hydrolase activity and proteins associated to histone modification and methylation (Appendix A), and Trinity might favor the prediction of proteins associated to nervous system processes (Appendix A). It is worth mentioning that one of the major drawbacks of our study is the lack of multiple venom gland samples; therefore, the previous findings might only be applicable to our dataset. Comparison to the previous *P. verdolaga* annotation was not possible due to major differences in the formatting of the gene ontology data.

To date, Trinity is regarded as the best de novo assembler [39,54]. However, it has been reported that the merging of various assemblers can further improve the quality of the annotation of a transcriptome [45]; thus, merging between the common and unique ORFs for all software was created and annotated (Appendix A). As observed with the individual software, the number of annotated proteins was highly limited by the BLAST hits. Merging of all ORFs introduced noise to the annotation, as a reduction of approximately 6% was observed in the ratio between sequences with annotation and without it. The biggest contributor to the percentage of sequences without annotation was SPAdes with 37%, while the SOAPdenovo-Trans annotation contributed the least with 7% (Table 4). Nonetheless, refraining from merging the Trinity and SOAPdenovo-Trans ORFs to the SPAdes unigenes could have avoided the annotation of approximately 1700 non-redundant genes, meaning a 18% reduction in the final number of annotated genes (Table 4). In total numbers and regarding GO-terms, analysis of the annotation of the merged common and unique ORFs represented: (i) an increase of 23% in the number of annotated proteins when compared to Trinity, (ii) an increase of 30% in the number of genes associated to molecular function GO-terms when compared to Trinity, (iii) an increase of 7% in the number of genes associated to biological process GO-terms when compared to Trinity, (iv) a reduction of 2% in the number of genes associated to the cellular component when compared to Trinity, and (v) the loss of representation of a total of ten GO-terms, which are related to 1197 molecular function, 6046 cellular component, and 8357 biological process genes, which accounts for approximately 5% of all genes related to the Trinity assembly (Table 5, Appendix A, Appendix A). In summary, the merging of the three assembled transcriptomes increased the total number of annotated proteins by 23%, but an approximate 5% of the information was lost. The CD-HIT software makes global alignments of all the sequences within a file, then it creates clusters of these sequences at a certain homology threshold, and finally reports the longest representative sequence for each cluster [55]. The clustering at an identity level of 85% reduced the total number of redundant structurally annotated ORFs from 42,029 to a total of 18,833 non-redundant ORFs (Table 2 and Appendix A), in which many valuable isoforms were surely lost. This could explain the general reduction of GO-terms and their associated genes, as observed within the merged annotation. Then, there is a clear need for the optimization of parameters such as k-mer length, CD-HIT homology thresholds, and inclusion criteria for the merging of the ORFS, as to attain the maximum potential this methodology could provide. However, as multiple transcriptomes from different biological samples would be required, such optimization is beyond the scope of this work. Nonetheless, this work presents a valuable approach using a merged assembly. Our findings highlight the importance of comparing different assembly software and carefully evaluating the resulting functional annotations to maximize the chance to identify proteins and peptides of biological interest.

### 3.3. New Toxins in the Venom Gland Transcriptome of P. verdolaga

Hidden Markov models, unlike local alignment methods such as BLAST, enable the recognition of a pattern within the parsed sequence based on statistical models [24]. In the case of spiders, this allows the identification of toxins by prioritizing important characteristics such as the cysteine scaffold. The software hmmcompete is a specialized tool that uses over 1000 spider sequences from the UniProtKB ToxProt database and creates HMMs that allow the discrimination of sequences into 219 spider toxin families based on various features, such as the cysteine pattern [35]. The prediction of the peptides/proteins with probable biotechnological interest was carried then with both local alignment and HMM prediction strategies with the software BLAST and hmmcompete, respectively. The HMM strategy allowed the identification of 186 unique ORFs of interest, while the BLAST strategy allowed the identification of 27 unique ORFs, thus confirming both strategies as complimentary.

All 328 hypothetical toxins were subjected to mature toxin prediction. Spider toxins usually contain an N-terminal signal peptide sequence, followed by a pro-peptide region of variable length, and a C-terminal cysteine-rich region that is termed as the mature toxin [56]. In total, 39% of *P. verdolaga’s* hypothetical toxins showed signal peptide presence, 66% showed pro-peptide cleavage sites, and 20% showed both conditions simultaneously. Therefore, only 65 sequences were reported as mature (Appendix A). The mature sequences, which were named U-theraphotoxin-Pv5a to U-theraphotoxin-Pv61a, showed lengths between 35 and 993 amino acids and 2 up to 56 cysteines, indicating the presence of both hypothetical small toxins and hypothetical proteins. In total, 15 of the 65 mature sequences presented odd cysteine numbers: between 3 and 19. Previously described *P. verdolaga* toxins [33], as well as several *Selenotypus plumipes* toxins [56], have been reported with odd cysteine numbers. Furthermore, odd cysteine numbers have also been described previously in cones [57]. Even when not usually described in toxins, unpaired cysteines play a role in other processes such as the polymerization of immunoglobulins [58], the stabilization of oxidized proteins [59], and in signal transduction associated to the sensing of reactive oxygen species [60], making them not completely uncommon. 

Available bioinformatic tools for the prediction of signal peptides are widely used for the processing of toxins; thus, the absence of a signal peptide in the precursor sequences could be an indicator that the protein is not secreted. However, unlike signal peptide prediction, pro-peptide determination is complex, as (i) not all spider toxins, such as CsTx-20, 21, and 22 from *Cupennius salei*, require containing a pro-peptide region in the toxin precursor [61], (ii) spider proprotein convertases (PPCs) favor the use of different recognition sites, such as the processing quadruplet motif, in comparison to the di-basic recognition motif [2], and (iii) available bioinformatic tools are mainly based on di-basic motif prediction and thus are only capable of correctly predicting between 10 and 80% of spider pro-peptide cleavage sites [56]. Consequently, there is an express need for the development of specialized tools that allow automated prediction of spider toxins.

The 65 mature hypothetical toxins as well as the 263 precursor toxins were grouped by hmmcompete into 77 toxin families (Appendix A). It is worth mentioning that the existence of homology between the toxins mentioned below and the hypothetical *P. verdolaga* toxins does not confirm their biological function, which needs to be confirmed in the wet lab. Thirteen toxins were classified as latarcin homologs, a group of approximately 12 alpha-helical antimicrobial peptides was described from the venom of *Lachesana tarabaevi*, and two toxins were classified as cupiennin homologs, antimicrobial peptides described from the venom of *Cupiennius salei*. Both latarcins and cupiennins show antimicrobial activity in both Gram-positive and Gram-negative bacteria in the sub-micromolar range with low cytotoxic activity [2,62,63]. However, cupiennins also possess immunomodulatory potential as they can regulate nitric oxide formation [64]. Nine toxins were classified as several atracotoxin (AcTx) homologs. AcTxs are described in the venom of spiders of the genera *Hadronyche* or *Atrax* and are mainly characterized by their insecticidal activity due to their affinity to insect Cav and voltage-gated Na+ channels [65,66]. Approximately 25% of the hypothetical *P. verdolaga* toxins were homologs to *L. tredecimguttatus* toxins. In total, 18 and 10 toxins were classified as α or δ-latroinsectotoxin homologs, respectively. These toxins induce neurotransmitter release only in arthropods, which could make their *P. verdolaga* homologs hypothetical insecticidal proteins [67,68]. Sixteen toxins were classified as α-latrotoxin homologs. Two main latrotoxins are described: α-Latrotoxin-Lt1a (*L. tredecimguttatus*) and α-latrotoxin-Lha1a from *Latrodectus hasselti*. α-latrotoxin-Lt1a is a 150 kDa toxin that shares 93% of its amino acid sequence with the toxin α-latrotoxin-Lha1a. Both α-latrotoxins show toxicity to vertebrates and in some situations can represent a fatal danger to mammals due to the formation of membrane pores that lead to neurotransmitter release [69]; still, the pore formation activity could be exploited in the development of various cytolytic drugs that may prove to be either antimicrobial or anti-cancerous. Thirty-eight hypothetical toxins were classified as α-latrocrustotoxin homologs, which affect crustaceans by promoting the release of neurotransmitters by various mechanisms [67]. However, it is known that this toxin is responsible for some of the most deleterious effects of black widow envenomation [70]. Furthermore, two toxins were classified as phospholipase A2 homologs and five as angiotensin-converting enzyme homologs. These two hypothetical phospholipases show homology smaller than 80% when compared to any of the previously reported phospholipase-D-verdolaga [33]. The identification of these hypothetical toxins might indicate that the bite from *P. verdolaga* could lead to edema, extreme pain, and possible necrosis on vertebrates. Twenty-three of the *P. verdolaga* toxins were classified as tachylectin homologs. Spider tachylectins are homologs to the toxin techylectin-5B, an antimicrobial lectin from the horseshoe crab *Tachypleus tridentatus* [71] that binds N-acetylglucosamine and N-acetylgalactosamine. Furthermore, as lectins have gained importance in the fields of immunology and glycobiology, these toxins might also prove to be valuable tools for research [72,73]. Proteins that might be related to the *P. verdolaga* toxin formation process were also described, since 28 protein disulfide isomerases, 27 serine proteases, one signal peptidase, and three cystatin homolog toxins were observed. Finally, five toxins were classified as peptidylglycine alpha-amidating mono-oxygenase (PAM) homologs, a group of enzymes dedicated to the biosynthesis of many signaling peptides [74]. To the best of our knowledge, these are the first reported PAM homolog sequences in spiders of the *Pamphobeteus* genera.

## 4. Conclusions

Here, the utilization of three distinct assembly algorithms and the merging of each of their non-redundant outputs allowed the identification and annotation of over 2000 more proteins when compared to the v1.0 of *P. verdolaga’s* transcriptome assembly previously performed in our group (over 28% improvement). This supports the need for the merging of the output of various assemblers for obtaining more complete transcriptomes. It is also worth mentioning that, as limited biological samples were available, the true optimization of variables such as read pre-processing, k-mer length, structural annotation settings, and optimal homology percentage for redundancy analysis, as well as quantitative criteria for ORF merging, was not feasible, and as a result an estimate of 5% of all gene information was lost during the merging process. Finally, the 9232 annotated proteins will serve as guidance for the description and annotation of posterior spider proteins, especially in the absence of nucleotide sequences of new world theraphosids. The prospection of bioactive peptides allowed the identification of 65 novel mature and 263 novel precursor theraphotoxins with interesting hypothetic activities that range from probable new antimicrobials to novel insecticidal proteins that may have a place in the development of new products in the midterm. These activities are then left to be tested biologically.

## 5. Materials and Methods

### 5.1. Venom Gland Transcriptome Data

The transcriptome’s raw reads from the venom gland of *Pamphobeteus verdolaga* were obtained from the European Nucleotide Archive (ENA) under accessions numbers: PRJEB21288/ERS1788422/ERX2067777-ERR2008012. As explained by Estrada and co-workers [32], two female specimens were collected from the province of Antioquia, Colombia, under the contract 155 signed by the University of Antioquia and the Environmental Ministry of Colombia, and the venom glands were extirpated. The total RNA was obtained through TRIzol^®^ reagent (ThermoFisher Scientific, Waltham, MA, USA) while the purification process of mRNA and the library creation was carried out with the Illumina mRNA TruSeq kit v2, as indicated by the manufacturer. The library 100 bp pair-ended reads was sequenced in an Illumina Hiseq 2500 instrument (Illumina Inc., San Diego, CA, USA).

### 5.2. Transcriptome Assembly and Statistics

The raw reads obtained from the Illumina platform were filtered of low-quality reads and possible adaptor sequences with the software TrimGalore v0.6.3 (https://github.com/FelixKrueger/TrimGalore, accessed on 31 July 2020) with default options. The cleaned reads were subject to an assembly with the software Trinity v2.1.1 (https://github.com/trinityrnaseq/trinityrnaseq, accessed on 31 July 2020) [75], SOAPdenovo-Trans v1.0.4 (https://github.com/aquaskyline/SOAPdenovo-Trans, accessed on 31 July 2020) [76], and the rnaspades.py pipeline from SPAdes v3.13.1 (https://github.com/ablab/spades, accessed on 31 July 2020) [77]. For the assembly, the only parameter modified was that of the k-mer length. The used lengths were 25 (default) and 32 (maximum) for Trinity, while k-mer lengths 31 (minimum, used for comparison with Trinity) and 63 (evaluation of higher k-mer lengths and effect on variability of predicted genes) were used for SPAdes and SOAPdenovo-Trans. No higher k-mer lengths were used for SPAdes and SOAPdenovo-Trans, as k-mer lengths close to the read length could drastically limit the amount of contigs assembled [78]. All outputs from the assemblies were analyzed for basic alignment quality with the software QUAST v5.0.2 (https://github.com/ablab/quast, accessed on 31 July 2020), as there are compatibility issues between the TrinityStats.pl script and the outputs from SPAdes and SOAPdenovo-Trans. Assessment of the read representation was carried out with the software Bowtie2 v.2.2.5 (https://github.com/BenLangmead/bowtie2, accessed on 31 July 2020) with a maximum number of reported reads of 20; the other parameters were set as default. Only the overall alignment rate was reported. The assembled contigs were aligned to the genome of the common house spider *Parasteatoda tepitadorium* as to assess the level of coverage. The alignment was performed with the software minimap2 v2.17 (https://github.com/lh3/minimap2, accessed on 31 July 2020) using the splice setting. 

### 5.3. Assessing Transcriptome Completion with BUSCO

A first measure of the completeness of the transcriptomes was evaluated using the Benchmarking Universal Single-Copy Orthologs (BUSCO) v4.1.2 software (https://github.com/WenchaoLin/BUSCO-Mod, accessed on 31 July 2020) [79,80]. For this, each of the assemblies were analyzed against BUSCO’s own Arthropoda dataset. The data from the number of complete, duplicated, fragmented, and missing BUSCO terms were extracted. In the case of the assemblies obtained from SOAPdenovo-Trans and SPAdes, a separation of transcripts with lengths higher and lower than 200 bp was carried out with the software SeqKit v0.12.0 (https://github.com/shenwei356/seqkit, accessed on 31 July 2020) [81] before BUSCO. 

### 5.4. Structural and Functional Annotation of the Transcriptome Assemblies

The transcriptome assemblies from Trinity, SPAdes, and SOAPdenovo-Trans that showed the best performance were annotated with the software Augustus v 3.3.3 (https://github.com/Gaius-Augustus/Augustus, accessed on 31 July 2020) [82,83] for the prediction of ORFs and with the software OmicsBox v2.0.36 (BioBam Bioinformatics S.L., Valencia, Spain) (previously known as BLAST2GO) for the functional annotation. Briefly, the structural annotation with Augustus was performed in intronless mode and used the genome of *P. tepitadorium* as a template in the generation of the Generalized Hidden Markov Models for predicting the possible ORFs from each of the assemblies. Only the longest complete ORF was reported. The intra and inter assembly redundancy analyses were carried out using the software CD-HIT v4.8.1 (https://github.com/weizhongli/cdhit, accessed on 31 July 2020) in default and 2D mode, respectively, considering a cutoff of 85% homology. The CD-HIT-2D used for comparison between assemblies was performed at least twice with all sequences acting both as query and subject. The merging of all transcriptomes was performed manually using the output from the CD-HIT-2D analysis. The functional annotation was carried out with the software OmicsBox using the UniProtKB database (release 2020_06) for homology search as well as the InterPro database (Release 80.0) for GO-terms retrieval. Only those sequences with hits from both databases were considered as annotated. All GO-terms for each annotation and their associated genes were extracted from the combined graph functionality. The GO-term analysis was carried out with the help of the Revigo software (available at http://revigo.irb.hr, accessed on 28 February 2022) with default options [84]. The resulting numbers for the structural annotation as well as the heatmap used for the GO-terms analysis were graphed with the software GraphPad Prism v9.0.1 for Mac OS (GraphPad Software, La Jolla, CA, USA).

### 5.5. Prediction of Proteins and Enzymes

Two strategies were used for the prospection of sequences of interest within the transcriptomes. The BLAST strategy used a total of 1027 sequences from the ArachnoServer database (UniProtKB—ArachnoServer, version 2022_02, accessed on 31 May 2022) matching to the keywords antimicrobial, antinociceptive, antiparasitic, antiarrhythmic, cytolytic, hemolytic, hyaluronidase, pro-inflammatory, kinin, lectin, necrosis, neurotoxin, neurotransmitter hydrolysis, presynaptic neurotoxin, neurotoxin, protease inhibitor, and protease; in addition, 608 sequences from the 90% homology cluster of the *Araneae* subset from the Animal Toxin Annotation Project (ToxProt) (UniProtKB-ToxProt, version 2022_02, accessed on 31 May 2022) were extracted from UniProtKB. These sequences were contrasted to all the non-redundant ORFs using the software BLAST+ v.2.10.0 [85] with a maximum e-value of 1 × 10^−5^. The HMM strategy used the software hmmcompete in its only released version with default options [35]. All merged annotation results were united into a single file, organized based on the ID given within the merged annotation, and finally manually curated. The prediction of the mature sequence was carried out as follows: (i) Each sequence was checked for the presence of signal peptide with SignalP 6.0 (https://services.healthtech.dtu.dk/service.php?ProP-1.0, accessed on 31 May 2022) [86] and the Phobius server (https://phobius.sbc.su.se, accessed on 31 May 2022) [87]. (ii) If a signal peptide was identified, the section coordinates were registered, and the associated N-terminal amino acid sequence was eliminated from the precursor ORF sequence. (iii) Pro-peptide prediction was carried with the ConoPrec (http://www.conoserver.org/?page=conoprec, accessed on 31 May 2022) [88] and ProP 1.0 (https://services.healthtech.dtu.dk/service.php?ProP-1.0, accessed on 31 May 2022) [89] servers using the sequences without signal peptide. (iv) If pro-peptide cleavage sites were identified, the section coordinates were registered, and the associated peptides were processed from the precursor sequence. (v) An alignment was carried between resulting sequences from steps ii, ii, and iv, along with the BLAST and/or hmmcompete homolog sequences. (vi) The proposed mature sequence was that in which the BLAST and/or hmmcompete homolog sequences were bounded by pro-peptide cleavage sequences. (vii) The sequences were classified either by the presence or absence of signal peptide and pro-peptide. (viii) The number of cysteines was calculated. A full summary of the prediction process can be observed in Figure 5.

## Figures and Tables

**Figure 1 toxins-14-00408-f001:**
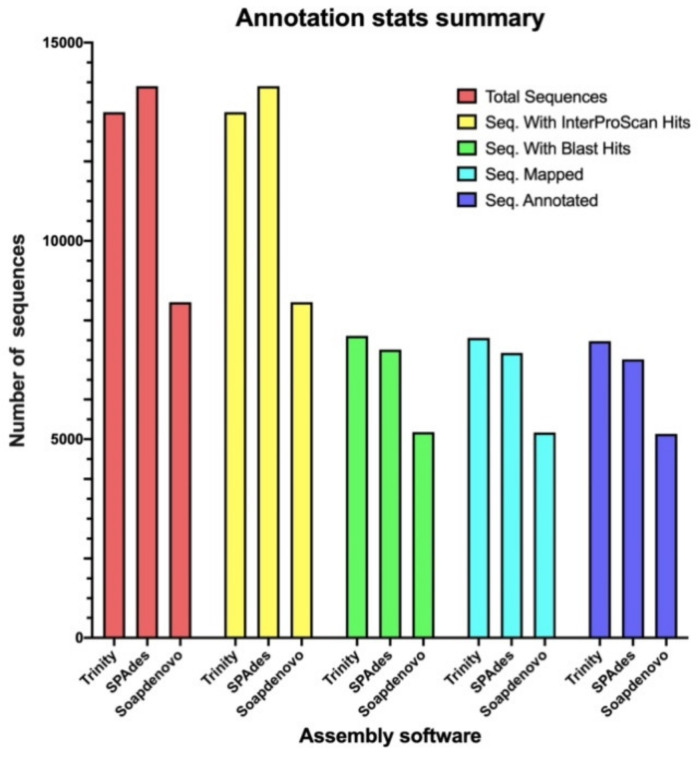
Summary of the functional annotation of the assembled transcripts from Trinity k-mer 25, SPAdes k-mer 31, and SOAPdenovo-Trans k-mer 63. From left to right: total number of ORFs assembled per each software, number of sequences with hits associated to InterProScan, number sequences with hits associated to BLAST, number of sequences with mapped InterProScan, and BLAST hits and number of annotated sequences.

**Figure 2 toxins-14-00408-f002:**
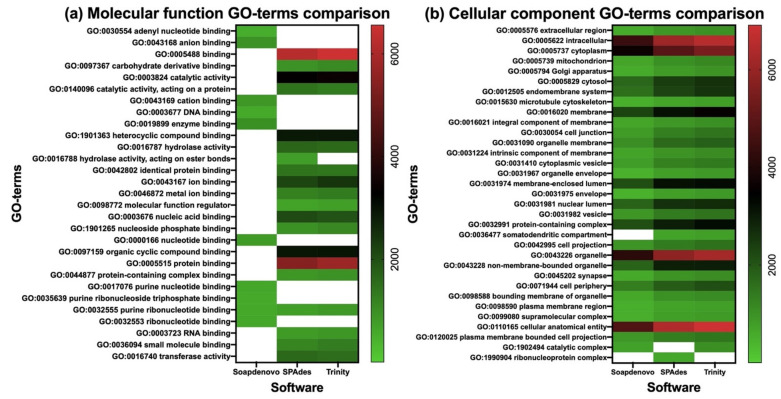
Comparison of the molecular function (**a**) and cellular component (**b**) non-redundant GO-terms associated to the functional annotation of the ORFs obtained from the SOAPdenovo-Trans k-mer 63, SPAdes k-mer 31, and Trinity k-mer 25 assemblies. Blank spaces represent absence of genes associated to the non-redundant GO-term.

**Figure 3 toxins-14-00408-f003:**
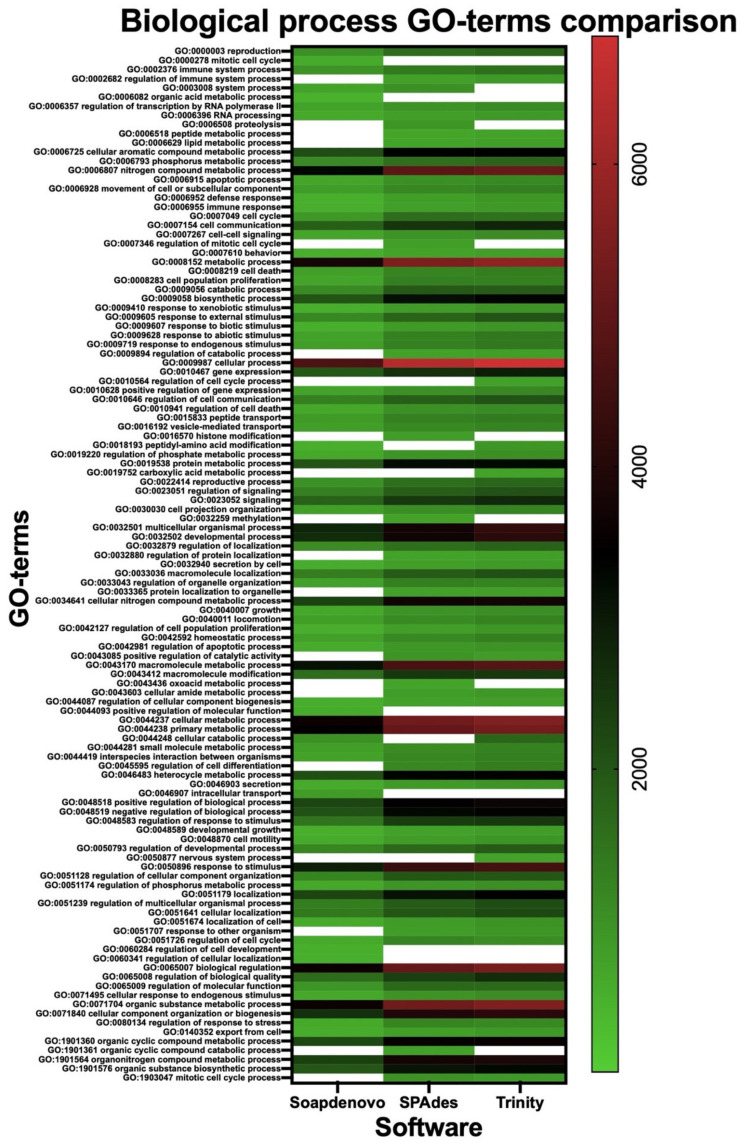
Comparison of the biological process non-redundant GO-terms associated to the functional annotation of the ORFs obtained from the SOAPdenovo-Trans k-mer 63, SPAdes k-mer 31, and Trinity k-mer 25 assemblies. Blank spaces represent absence of genes associated to the non-redundant GO-term.

**Figure 4 toxins-14-00408-f004:**
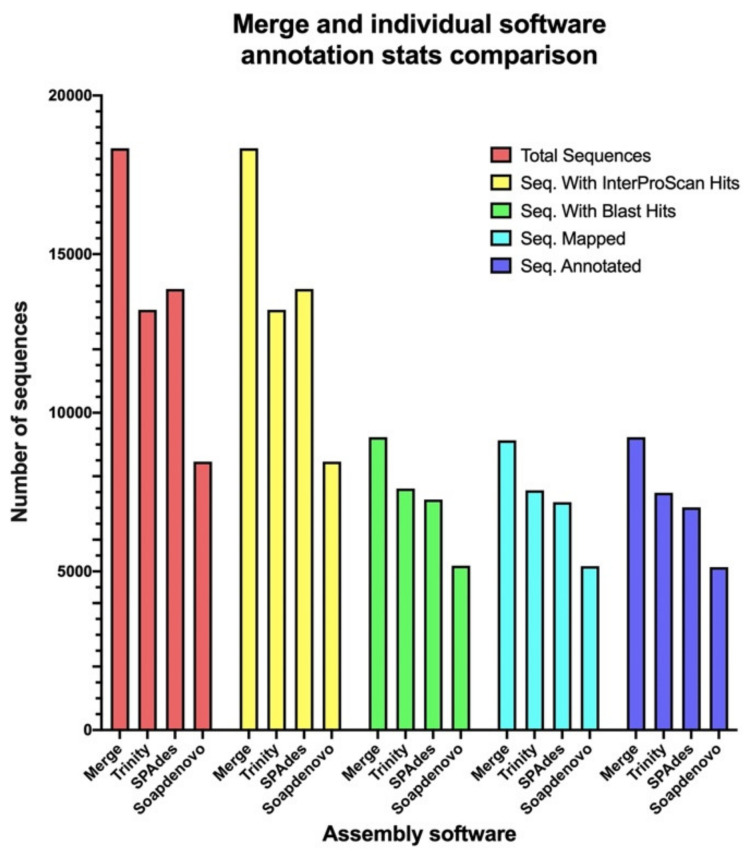
Comparison of the functional annotation stats of the assembled transcripts from Trinity k-mer 25, SPAdes k-mer 31, SOAPdenovo-Trans k-mer 63, and the merging of the unique and common ORFs between all software. From left to right: total number of ORFs assembled per each software, number of sequences with hits associated to InterProScan, number sequences with hits associated to BLAST, number of sequences with mapped InterProScan and BLAST hits, and number of annotated sequences.

**Figure 5 toxins-14-00408-f005:**
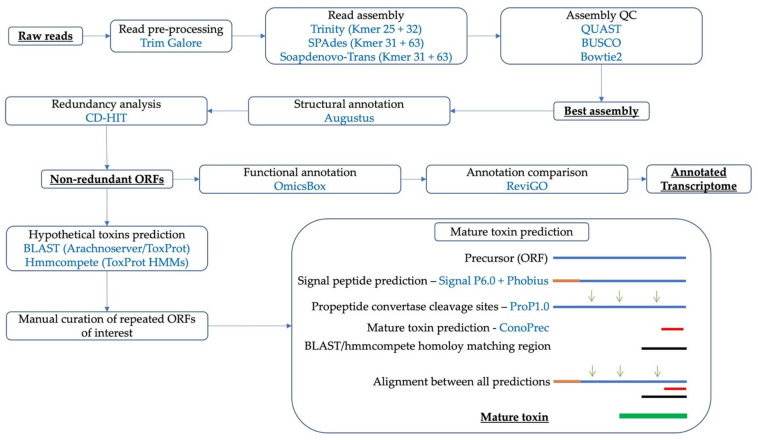
Graphical summary of the assembly, ORFs of interest prediction, and mature peptide prediction methodologies.

**Table 1 toxins-14-00408-t001:** Summary of the stats associated with the assembly of the 24 million raw reads from the venom gland transcriptome of *P. verdolaga* as obtained with the software Trinity, SPAdes, and SOAPdenovo-Trans at k-mer lengths between 25 and 63. At the top, the summary of the Quast analysis and the read representation analysis carried out with Bowtie2. At the middle is a summary of the *P. tepitadorium* alignment (al.) representation analysis. At the bottom is a summary of the BUSCO analysis.

	Trinity k-mer 25	Trinity k-mer 32	SPAdes k-mer 31		SPAdes k-mer 63		SOAPdenovo-Trans k-mer 31		SOAPdenovo-Trans k-mer 63	
Number of contigs > 200 bp	99,316	96,821	80,030		81,831		68,700		59,910	
Number of contigs < 200 bp	0	0	58,250		32,694		1,348,391		169,055	
%GC	39.5	39.82	39.42		39.89		38.93		40.21	
N50	772	797	882		722		468		504	
Number of nt	56,851,075	56,627,550	49,124,765		44,839,329		29,425,929		26,911,362	
Raw read representation in contigs	95.68%	96.82%	96.94%		94.72%		16.66%		92.47%	
*P. tepitadorium* al. average length	14,083	11,069	11,269		7819		7091		5050	
*P. tepitadorium* al. total coverage	29,236,516	25,625,827	17,635,778		15,521,127		3,148,352		6,004,386	
*P. tepitadorium* al. average quality	14	12	15		11		20		11	
*P. tepitadorium* al. max length	247,989	247,989	247,989		247,989		106,194		168,652	
*P. tepitadorium* al. min length	40	40	40		40		40		40	
Size	>200 bp	>200 bp	<200 bp	>200 bp	<200 bp	>200 bp	<200 bp	>200 bp	<200 bp	>200 bp
Complete BUSCOs	79.36%	77.95%	0%	79.36%	0%	74.01%	0%	46.15%	0.1%	58.26%
Fragmented BUSCOs	9.1%	9.47%	3.65%	9.01%	3.28%	12.76%	16.51%	32.18%	6.38%	22.42%
Missing BUSCOs	11.53%	12.57%	96.34%	11.63%	96.72%	13.23%	83.49%	21.67%	93.52%	19.32%
Duplicated BUSCOs	24.23%	23.70%	0%	8.98%	0%	6.46%	0%	1.42%	0%	2.25%

**Table 2 toxins-14-00408-t002:** Summary of the structural annotation stats for the assembled transcripts from Trinity k-mer 25, SPAdes k-mer 31, and SOAPdenovo-Trans k-mer 63. The stats associated to the individual (redundancy) and collective (uniqueness) CD-HIT analysis for each assembly are also shown.

	Number of ORFs	Average Length (aa)	Maximum Length (aa)	Total Amino Acid Number	% Redundant ORFs	Number of nr ORFs	Number of Uniquely Assembled ORFs
Trinity k-mer 25	17,706	287.5	4077	5,089,960	25.20	13,244	5112
SPAdes k-mer 31	15,858	239.3	3777	3,795,493	12.31	13,905	5085
SOAPdenovo-Trans k-mer 63	8465	217.7	1790	1,842,643	0.07	8459	942

**Table 3 toxins-14-00408-t003:** Summary of the analysis of the GO-terms associated to the annotated proteins in the SOAPdenovo-Trans k-mer 63, SPAdes k-mer 31, and Trinity k-mer 25 assemblies. The total number of genes associated to each GO-term category is also shown.

GO-Term		SOAPdenovo-Trans		SPAdes		Trinity	
Category	Total Terms	Identified Terms	Associated Genes	Identified Terms	Associated Genes	Identified Terms	Associated Genes
Molecular function	29	10	7086	20	39,266	19	39,986
Cellular component	33	31	45,010	32	64,844	32	71,026
Biological process	113	93	122,853	102	190,438	100	202,291

**Table 4 toxins-14-00408-t004:** Summary of the contribution of the unique and common ORFs between all software found in the total number of annotated sequences within the merged annotation.

	Merged File	Common ORFs	Unique Trinity ORFs	Unique SPAdes ORFs	Unique SOAPdenovo-Trans ORFs
Total sequences	18,338	7199	5112	5085	942
Annotated	9232	4552	2703	1695	282
Without annotation	9106	2647	2409	3390	660
Contribution to sequences without annotation	NA	29.07%	26.45%	37.23%	7.25%

**Table 5 toxins-14-00408-t005:** Summary of the analysis of the GO-terms associated to the annotation of the merged ORFs. The unique merged GO-terms and their relationship to the missing GO-terms present in the individual software annotation are also shown.

GO-Term		Merge			
Category	Total Terms	Identified Terms	Associated Genes	Unique Terms	Missing Terms
Molecular function	29	19	51,542	-	“GO:0098772 molecular function regulator”
				-	“GO:0032555 purine ribonucleotide binding”
Cellular component	33	31	69,249	-	“GO:0110165 cellular anatomical entity”
Biological process	118	97	216,509	-	“GO:0003008 system process”
				-	“GO:0006955 immune response”
				“GO:1901698 response to nitrogen compound”	“GO:0009410 response to xenobiotic stimulus”
				-	“GO:0015833 peptide transport”
				-	“GO:0018193 peptidyl-amino acid modification”
				“GO:0051049 regulation of transport”	“GO:0032880 regulation of protein localization”
				-	“GO:0032940 secretion by cell”
				“GO:0015031 protein transport”	“GO:0033365 protein localization to organelle”
				“GO:2000026 reg. of multicell. orga. develop.”	“GO:0045595 regulation of cell differentiation”
				-	“GO:0046903 secretion”
				-	“GO:0140352 export from cell”
				“GO:0044770 cell cycle phase transition”	“GO:1903047 mitotic cell cycle process”

**Table 6 toxins-14-00408-t006:** Summary of the screening of peptides and proteins of probable biotechnological interest, homolog to the reported proteins of the ArachnoServer database (Adb), and the *Araneae* subset of the Animal Toxin Annotation Project (ToxProt) using BLAST.

	ArachnoServer				ToxProt			
Associated Keyword	Trinity	SPAdes	SOAPdenovo-Trans	Merge	Trinity	SPAdes	SOAPdenovo-Trans	Merge
Antimicrobial	0	4	0	4	5	8	2	9
Antinociceptive	1	1	0	1	1	4	0	5
Antiparasitic	0	0	0	0	0	0	0	0
Antiarrhythmic	1	1	0	1	0	0	0	0
Cytolytic	2	2	2	2	0	0	0	0
Hemolytic	2	2	2	2	3	8	2	8
Hyaluronidase	1	1	0	1	0	0	0	0
Pro-inflammatory	1	1	0	1	2	2	4	2
Kinin	0	0	0	0	0	0	0	0
Lectin	17	22	8	24	0	4	0	4
Necrotic	2	2	2	2	2	2	2	2
Neurotoxin	19	22	4	26	89	78	47	108
Neurotransmitter hydrolysis	3	4	1	4	0	0	0	0
Presynaptic neurotoxin	66	50	41	70	64	38	39	69
Protease inhibitor	13	9	4	14	12	9	4	13
Protease	22	18	15	22	0	0	0	0
Total	150	139	79	174	178	153	100	220

**Table 7 toxins-14-00408-t007:** Summary of the screening of peptides and proteins of probable biotechnological interest obtained from the HMM search strategy with the software hmmcompete.

hmmcompete’s HMMs	Trinity	SPAdes	SOAPdenovo-Trans	Merge
Cationic peptides	7	9	6	8
Neurotoxins	58	53	24	77
Venom proteins	221	180	132	233
Total	286	242	162	318

## Data Availability

All the assemblies have been deposited at DDBJ/EMBL/GenBank under the accession GIUY00000000. The version described in this paper is the first version, GIUY00000000. All hypothetical mature toxins have been deposited at the EMBL-EBI European Nucleotide Archive under accession numbers OX043992 to OX044319.

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
