# Peer review of "Improving the Annotation of the Venom Gland Transcriptome of Pamphobeteus verdolaga, Prospecting Novel Bioactive Peptides"

_toxins, 2022, doi:10.3390/toxins14060408_

Round 1

Reviewer 1 Report

This manuscript deals with a relevant analysis of venomous animals, specially focus at the trancriptome analysis of venom glands.  However, there are a few points that need revision.

Major points

  1. Since one of the objectives is to recover better data on sequences and annotations, it is suggested that the authors use the Animal Toxin Annotation Project of UniProt, which has more than 7000 proteins of venoms (1,569 of Araneae), whereas arachnoserver has only 1.164 proteins annotated.
  2. The authors say that the ensemble obtained with SPADES and SOAPdenovo permitted to identify sequences with less than 200 pb presented by ortologs. Thus, I guess you could compare these data by processing short sequences also with Trinity.
  3. For the sequences in which you could not identify signal peptides with the software SignalP, you could try to use the Phobius software (InterproScan) for prediction of the missing sites.

Minor points

  1. In section 2.1 it says that you used TrimGalore for cleaning the reads, but in section 5.2 you say to have used Trimmomatic. Pease clarify which was used (both?) and justify.
  2. In section 3.3 there is an incorrect affirmation: Hyaluronidase of Loxosceles is not dermonecrotic. It is a “spreading factor” that facilitates the dispersion of phospholipases into the tissues. The dermonecrotic effects of these spiders are caused by a Phospholipase D.
  3. In section 5.4 it says to have used a pair of values of K-mer of each ensemble. Did you assay also other values of K.mer? Having used only two K-mer for each ensemble is in my opinion rather poor decision. The objectives were to use the best configuration for obtaining more information.
  4. In section 5.4, the authors say they determined ExN50, but in the entire manuscript they mention only N50. You should use always the same definition.

Reviewer 2 Report

The article titled “Improving the annotation of the venom gland transcriptome of Pamphobeteus verdolaga, prospecting novel bioactive peptides” demonstrates a valuable approach in re-assembling the transcriptome of the the spider venom gland, using the combination of the three algorithms: Trinity, Soapdenovo and SPAdes. This resulted in the identification of 133 novel polypeptides of biological interest.

The manuscript is clearly written and does not exhibit controversies. Nevertheless, it can be improved in several aspects.

1. In the Introduction section authors state: “Interestingly, transformed tumor cells, also rich in PG, are targeted by NDBPs…”. The tumor cells feature the presence of phosphatidylserine (PS), not PG.

2. The interesting findings of the work are presented in the Table S2. These are mature polypeptides, identified with the new approach. However, some of the peptides do not look as mature ones, because they contain uneven number of Cys-residues. E.g. Pv8a (3 Cys), Pv35a (1 Cys), Pv36a (5 Cys). May be, this is the result of wrong gene assembly?

3. Table S2 is based on the annotation, using the Arachnoserver database. As a result Hypothetical activities (column 2) looks strange. E.g. Pv135a exhibits high homology (~74%) to the tarantula toxin jingzhaotoxin-XI (κ-theraphotoxin-Cj1a), which regulates the activation and inactivation of the voltage-gated sodium channel Nav1.5. Thus, this toxin is neurotoxin? Why in the Table S2 it is indicated that this toxin, is Antimicrobial, lectin, neurotoxin, protease inhibitor? Also, Pv35a and Pv36a look very similar but the first is simply Neurotoxin but the second features Antimicrobial, lectin, neurotoxin, protease inhibitor -activities. From my opinion, information in the column2 could be updated for a number of toxins if BLAST search against uniprot.org database could be performed.

4. Table S2 is difficult to accept in its current form. Is it possible to re-group the toxins and start from shortest toxins and move to longest (or group them, according to homology)? In addition, if the length of the toxins and number of Cys-residues would be provided, this solved the problem.

Date of this review

5 April 2022 16:20

Reviewer 3 Report

In this article, authors propose a comparison between the re-assembly of transcriptomic reads from a spider venom gland using different assembly tools. They state that the combination of assemblers (at least Two of them) is recommended to obtain a better annotation of the transcriptome content.

In the principle, the obtained result could have been hypothesized since the reason to combine tools is to take additive advantage from the “best of both”. This study has the merit to test this and establish it in a systematic approach. The impressive number of claimed new peptide identification is another reason to investigate the proposed approach and probably re-analyze available data.

However, article cannot be accepted in the current form and requires additional analyses with up to date tools.

Round 2

Reviewer 3 Report

It seems that authors did not received / considerd the major concerns raised from the previous version(https://susy.mdpi.com/user/review/displayFile/25794293/Rfsm4XHE?file=review&report=18772116). 

In fact they only answered to Point 1, asking for suggestions already provided in the previous report.

We report hereafter the points that really represent No-Go .

  1. The introduction should state current annotation procedure and briefly explain why a combination of assemblers could be an advantage.
  2. The annotation methodology is not clearly state and easy to follow. Authors should propose a graphicla representation of what they did and when/where each tool was used. This will allow and facilitate their result validation and repeatability.
  3. Genome assemblers are used in comparison with transcriptome assemblers.
  4. Do authors suggest that a transcriptome could contain introns?
  5. Authors shoud use a more recent version of Trinty. The one used is out of date since a decade.
  6. A GO-Based annotation is not relevant enough. Author should present results of BLAST ou HMM-Based analyses
  7. Authors must explain the choise of different k-mer length in the annotation procedure.

Author Response

Greetings and thank you very much for your feedback we appreciate it. First and foremost, we would like to apologize as we were not aware of the comments associated with the pdf document since in the platform we were not able to see a “please see the attached file”, and no apparent download link was observed. Once clarified our mistake, we proceed to answer the first 20 points included in the pdf file since these allow us to provide further information when compared to the 7 points mentioned in the second phase of the review process in the attached file below.

Round 3

Reviewer 3 Report

  1. Authors have argued a lot against points raised after the initial revue and have justified their inital choices.
  2. They have recognized their  "error" of using genomic tools for analysis of transcriptomic data and have provided  command lines with the correct tools to be used. It is however unclear if they relaunched the analysis with the latter commands since very strangely results are similar to the one proposed in version 1. It seems that the only update concerns the name of the tool. the only change concerns the number of novel theraphotoxin identified which is 74 since version 2. the number 74 was proposed in version 2 without any explanation and remains unchanged after the supposed change of command lines
  3. Authors have stated in their answer that they will not modify their approch since using the proposed HMM-based approach will be too long and would imply too many changes in their manuscript. We recommand to take the appropriate time to go through this approach demonstrated to provide more confident and useful results.
  4. The newly identified sequences are not deposited in genbank. This is a prerequisite however, authors propose to do that later paper acceptation.
  5. We asked to include in the manuscript a figure of the pipeline with a clear succession of tools used for the analysis.  This figure is not provided.
  6. A number of recent publications concerning spider transcriptomic  sequence annotation were not considered to state the current methodology and justify clearly author choices (Go-based annotation). This has been provided in authors response and should be also included eitheir in the introduction or the discussion of the manuscript.

Round 4

Reviewer 3 Report

We thank authors for the high level of improvement observed since the previous version. As expected, they have kept the BLAST based annotation with somehow useless figures. But, also as expected, the HMM-based approach has highly improved the annotation. the number of toxin uniquely identified by the latter (186 unique sequences from the HMM strategy, 27 unique from the BLAST strategy)

However, some minor concerns are still to be considerd:

1. the usage of HMM-based annotation is not indicated in the introduction. It is however evident that this annotation is of importance in evaluating the assemblies and moreover is the key aspect of new peptide identification.

2. It is not clear what authors mean by "inmature toxin". Do they refer to "toxin precursor" or "fragment of the precursor not containing a mature peptide"? In any case, the prediction of the exact mature peptide is somehow difficult in fragment missing the Quatruplet processing motif (PQM) that is used by the protease to excise the final mature peptide from the precursor.

3. In the discussion, Authors seem to point out the function of identified toxins (from line 600). It is important to clarify that bioinformatic annotation is not able to predict a biological function. The proposed classification remains putative and more than hypothetical until verified with an experiment. We recommand authors to avoid referring to predicted toxin with "precise" pharmacological activities like alpha, delta or other.

4. The graphical abstract is not easily readable. Please provide a figure with better quality.

5. The claimed newly discovered sequence have to be made available prior to manuscript publication. it is now too common to see paper that claimed sequences of interest and never disclosed them...

Author Response

Greetings. Please see attached file.
